 

# Epigenome-wide analysis of DNA methylation and coronary heart disease: a nested case-control study

Jiahui Si[1,2], Songchun Yang[1], Dianjianyi Sun[1], Canqing Yu[1], Yu Guo[3], Yifei Lin[4], Iona Y Millwood[5,6], Robin G Walters[5,6], Ling Yang[5,6], Yiping Chen[5,6], Huaidong Du[5,6], Yujie Hua[7], Jingchao Liu[8], Junshi Chen[9], Zhengming Chen[6], Wei Chen[10], Jun Lv[1,11,12]*, Liming Liang[2]*, Liming Li[1]*, China Kadoorie Biobank Collaborative Group

[1]Department of Epidemiology and Biostatistics, School of Public Health, Peking University Health Science Center, Beijing, China; [2]Departments of Epidemiology and Biostatistics, Harvard T.H. Chan School of Public Health, Boston, United States; [3]Chinese Academy of Medical Sciences, Beijing, China; [4]Department of Urology, West China Hospital, Sichuan University, Chengdu, China; [5]Medical Research Council Population Health Research Unit at the University of Oxford, Oxford, United Kingdom; [6]Clinical Trial Service Unit & Epidemiological Studies Unit (CTSU), Nuffield Department of Population Health, University of Oxford, Oxford, United Kingdom; [7]NCDs Prevention and Control Department, Suzhou CDC, Jiangsu, China; [8]NCDs Prevention and Control Department, Wuzhong CDC, Jiangsu, China; [9]China National Center for Food Safety Risk Assessment, Beijing, China; [10]Department of Epidemiology, School of Public Health and Tropical Medicine, Tulane University, New Orleans, United States; [11]Key Laboratory of Molecular Cardiovascular Sciences (Peking University), Ministry of Education, Beijing, China; [12]Peking University Institute of Environmental Medicine, Beijing, China

*For correspondence:
lvjun@bjmu.edu.cn (JL);
lliang@hsph.harvard.edu (LL);
lmlee@vip.163.com (LL)

## Abstract

**Background:** Identifying environmentally responsive genetic loci where DNA methylation is associated with coronary heart disease (CHD) may reveal novel pathways or therapeutic targets for CHD. We conducted the first prospective epigenome-wide analysis of DNA methylation in relation to incident CHD in the Asian population.

**Methods:** We did a nested case-control study comprising incident CHD cases and 1:1 matched controls who were identified from the 10 year follow-up of the China Kadoorie Biobank. Methylation level of baseline blood leukocyte DNA was measured by Infinium Methylation EPIC BeadChip. We performed the single cytosine-phosphate-guanine (CpG) site association analysis and network approach to identify CHD-associated CpG sites and co-methylation gene module.

**Results:** After quality control, 982 participants (mean age 50.1 years) were retained. Methylation level at 25 CpG sites across the genome was associated with incident CHD (genome-wide false discovery rate [FDR] < 0.05 or module-specific FDR < 0.01). One SD increase in methylation level of identified CpGs was associated with differences in CHD risk, ranging from a 47 % decrease to a 118 % increase. Mediation analyses revealed 28.5 % of the excessed CHD risk associated with smoking was mediated by methylation level at the promoter region of *ANKS1A* gene (P for mediation effect = 0.036). Methylation level at the promoter region of *SNX30* was associated with blood pressure and subsequent risk of CHD, with the mediating proportion to be 7.7 % (*P* = 0.003) via

systolic blood pressure and 6.4 % (*P* = 0.006) via diastolic blood pressure. Network analysis revealed a co-methylation module associated with CHD.

**Conclusions:** We identified novel blood methylation alterations associated with incident CHD in the Asian population and provided evidence of the possible role of epigenetic regulations in the smoking- and blood pressure-related pathways to CHD risk.

**Funding:** This work was supported by National Natural Science Foundation of China (81390544 and 91846303). The CKB baseline survey and the first re-survey were supported by a grant from the Kadoorie Charitable Foundation in Hong Kong. The long-term follow-up is supported by grants from the UK Wellcome Trust (202922/Z/16/Z, 088158/Z/09/Z, 104085/Z/14/Z), grant (2016YFC0900500, 2016YFC0900501, 2016YFC0900504, 2016YFC1303904) from the National Key R&D Program of China, and Chinese Ministry of Science and Technology (2011BAI09B01).

## Introduction

Coronary heart disease (CHD) is one of the leading causes of morbidity and mortality worldwide (*GBD 2017 Causes of Death Collaborators, 2018*). Despite known environmental risk factors and the identification of genetic variations, a considerable proportion of the observed CHD risk remains unexplained (*Deloukas et al., 2013*).

Methylation at cytosine-phosphate-guanine (CpG) dinucleotides is a common epigenetic modification of DNA (*Deaton and Bird, 2011*), which forms an interface between the genotype and the environment (*Rosa-Garrido et al., 2018*). DNA methylation are responsive to environmental stimuli and unhealthy lifestyles, including smoking (*McCartney et al., 2018*), alcohol consumption (*Liu et al., 2018*), and obesity (*Wahl et al., 2017*). This makes DNA methylation a potential biomarker of environmental-related and lifestyle-driven diseases of adulthood, for example, metabolic dysfunction (including hypertension (*Richard et al., 2017*), diabetes (*Chambers et al., 2015*), and atherogenic dyslipidemia (*Irvin et al., 2014*). Unhealthy lifestyles, together with metabolic dysfunction, will further increase the risk of cardiovascular disease. Investigating the environmentally responsive DNA methylation change linked to CHD could gain insights into the underlying mechanisms and identify novel clinical biomarkers and therapeutic targets of CHD.

Previous epigenome-wide analysis of DNA methylation and CHD was characterized by small sample size (*Silvio et al., 2014*; *Nakatochi et al., 2017*; *Guarrera et al., 2015*; *Sharma et al., 2014*; *Li et al., 2017*; *Yamada et al., 2014*), based in primarily Western countries (*Silvio et al., 2014*; *Guarrera et al., 2015*; *Yamada et al., 2014*; *Golareh et al., 2019*; *Liu et al., 2017*; *Fernández-Sanlés et al., 2018*; *Rask-Andersen et al., 2016*), focusing on selective genomic regions (*Guarrera et al., 2015*; *Sharma et al., 2014*), or the cross-sectional nature of findings which precludes establishment of any temporal relationship (*Silvio et al., 2014*; *Nakatochi et al., 2017*; *Sharma et al., 2014*; *Li et al., 2017*; *Yamada et al., 2014*; *Liu et al., 2017*; *Fernández-Sanlés et al., 2018*; *Rask-Andersen et al., 2016*). Only a few prospective studies were conducted in the white populations (*Guarrera et al., 2015*; *Golareh et al., 2019*).

We examined the association between epigenome-wide methylation of blood-derived DNA and CHD risk over the next 10 years, by comparing prospectively ascertained CHD cases with 1:1 matched controls in the China Kadoorie Biobank (CKB). We then examined the relationships between the identified CHD-associated methylation sites and cardiovascular risk factors, and further identified potential pathway by causal mediation analysis. The overall analysis flowchart is provided in *Figure 1*.

## Materials and methods
### Study population

The CKB is a prospective cohort of 512,715 adults aged 30–79 years from 10 geographically diverse areas across China (five urban and five rural areas) since 2004–2008. Details of the study design, survey methods, and long-term follow-up have been given elsewhere (*Chen et al., 2011*). Briefly, all participants completed laptop-based questionnaires (including sociodemographic, lifestyle factors, and medical and medication history) and physical measurements (including body weight, height, and blood pressure). Participants also provided a 10 ml random blood sample for an immediate on-site test of random plasma glucose and long-term storage, with the time since last meal recorded. Mortality



**Figure 1.** Flowchart of the present study. CHD = coronary heart disease; QC = quality control; CpG = cytosine-phosphate-guanine; FDR = false discovery rate.

and morbidity during follow-up were identified through linkage with local death and disease registries, with the national health insurance system, and by active follow-up if necessary (i.e., visiting local communities or directly contacting participants).

## Study design

Baseline DNA methylation was measured for 494 CHD cases, whose CHD occurred during the follow-up period until 31 December 2015, and 494 matched controls. All these participants were free of heart disease, stroke, or cancer at baseline. They also had clinical chemistry measured for baseline plasma sample, including total cholesterol (TC), low-density lipoprotein cholesterol (LDL-C), high-density lipoprotein cholesterol (HDL-C), and triglycerides (TG) (Wolfson Laboratory at University of Oxford, UK).

Incident CHD cases were defined as fatal ischemic heart disease (IHD) coded as ICD-10 I20-I25 and nonfatal acute myocardial infarction coded as I21. The diagnosis adjudication has finished for 134 reported cases by a review of hospital medical records. Overall, 90 % of the diagnoses of CHD were confirmed. Cases were excluded if they have developed malignant neoplasms (C00-C97) or cerebrovascular diseases (I60-I69) during follow-up. Each case was individually matched to one control who was free of IHD, malignant neoplasms, or cerebrovascular diseases throughout follow-up. Controls were matched to cases by birth year ( ± 3 years), age at baseline ( ± 3 years), sex, study area, hours fasting prior to blood draw (0- < 6, 6- < 8, 8- < 10, and ≥10 hours) at baseline.

## Measurement of DNA methylation

For 494 pairs of CHD cases and controls, epigenome-wide methylation level of baseline blood leukocyte DNA was measured by Infinium Methylation EPIC BeadChip (Illumina, USA), which interrogates ~850,000 CpG sites (BGI, China). Although the laboratory staff were blinded to case/control status, the cases and controls were not strictly randomized on arrays.

We used minfi package (RRID:SCR_012830) to process methylation data. CpG sites were excluded if they: (1) were assayed SNPs rather than CpGs (n = 59); (2) had bead count <3 in 5 % of samples (n = 1,644), or had >1% of samples with a detection $P > 0.05$ (n = 2,536); (3) were overlapped with SNPs in the 1,000 Genome Project (20130502 release) with minor allele frequency in Eastern Asian population >0.05 at the target CpGs sites, single base extension sites of Type I probe, or the probe body (*Pidsley et al., 2016*); (4) possibly cross-hybridized to other genomic locations (*Pidsley et al., 2016*) (3 and 4 contained 132,762 sites in total). Samples were excluded if they (1) were outliers detected by multidimensional scaling analysis (n = 0); (2) were sex mixed-up samples (n = 2); (3) had missing rate >0.01 across CpG sites (n = 2); (4) were measured in a distinct study batch (n = 2).

After quality control, 982 of 988 samples with 747,726 CpG sites were retained. Also, we randomly chose 11 samples (one sample per plate) for duplicate measurements. The correlation for duplicate measurements on the same sample ranged from 0.992 to 0.997.

## Assessment of covariates

In the baseline questionnaire, for smoking, we asked frequency, type, and amount of tobacco smoked per day for ever smokers, and reason to quit for former smokers. We included former smokers who stopped smoking for illness in the current smoker category to avoid misleadingly elevated risk. We then calculated the current average number of cigarette equivalents consumed per day. For alcohol consumption, we asked drinking frequency on a week, type of alcoholic beverage, and volume of alcohol consumed on a typical drinking day. We calculated average pure alcohol volume consumed per day. For physical activity, we asked the usual type and duration of activities. The daily level of physical activity was calculated by multiplying the metabolic equivalent tasks (METs) value for a particular type of physical activity by hours spent on that activity per day and summing the MET-hours for all activities. For dietary habit, we used a short qualitative food frequency questionnaire to assess habitual intakes of 12 conventional food groups, that are mainly addressed in the Chinese dietary guidelines (2016). We then calculated a diet score: consuming fresh vegetables and fruits every day, red meat <7 days/week, soybean products ≥ 4 days/week, fish ≥1 day/week, and coarse grains ≥ 4 days/week, each item as one score. We then summed the above six scores for the total diet score. Trained staff measured weight and height with calibrated instruments. Body mass index (BMI) was calculated as weight in kilograms divided by the square of the height in meters.

## Statistical analysis

### Single DNA methylation marker and incident CHD

In the epigenome-wide analysis, raw methylation matrices were normalized using the dasen method in the wateRmelon package (RRID:SCR_001296). Linear regression was applied for single-marker tests, with the beta-values of methylation as dependent variables, CHD as an indicator, and age (years), sex (male or female), 10 study area, fasting time (< 8 or ≥ 8 hr), education level (no formal or primary school, middle or high school, technical school or college or higher), marital status (married or not), smoking (current average number of cigarette equivalents consumed per day), alcohol consumption (average pure alcohol volume consumed per day), physical activity (MET-hours), diet score (continuous variable ranging from 0 to 6), and BMI ($kg/m^2$) as covariates. To quantify latent factors, including the effects of unobserved batch effects, cell compositions, and other unmeasured confounding factors, we used smart surrogate variable analysis by the smartSVA package (*Chen et al., 2017*). This method has been reported to be a fast and robust method for removing batch effects and preserve power (*Brägelmann and Lorenzo Bermejo, 2019*). Variables considered in the smart surrogate variable (SV) analysis included case or control status and all covariates. A total of 56 SVs were generated and also included as covariates in the above model. We used false discovery rate (FDR) < 0.05 to determine epigenome-wide significant CpGs in relation to CHD. We annotated CpGs to genes based on official EPIC array annotation file from *Illumina, 2017*.

In sensitivity analysis, we excluded 100 participants who reported usage of blood pressure-lowering drugs at baseline to avoid potential confounding effect caused by medications.

### Weighted gene co-methylation network and incident CHD

We also used the network approach to first identify CHD related co-methylation network module and then CHD related CpGs within the discovered module. We used weighted gene co-methylation network analysis (R package WGCNA, RRID:SCR_003302) to identify potential co-thylation network related to CHD. To ensure computation feasibility, we selected the top 20,000 CHD-associated CpGs from single-marker tests. This is about the maximum number of CpGs the WGCNA package can handle on our high performance computing cluster. Two samples were outliers and excluded during sample clustering. We used the function "blockwiseModules" with a minimum module size of 30 sites to construct network automatically. Modules were created and merged with the mergeCutHeight set to 0.25. We then identified modules that were statistically significantly associated with CHD using the module eigengene (the first principal component of the given module), with the same set of covariates as in the individual CpG association analysis. After detection of CHD associated modules, we performed the visualization of network modules and its hub gene to depict the connection among the annotated genes by VisANT 5.0 (http://visant.bu.edu/). Because the module was rather large, we restricted the genes used in the visualization to the annotated genes of the 24 CpGs with module-specific FDR < 0.01.

To ensure the selection of top 20,000 CpGs did not inflate the false positives of CHD-module association, we carried out a permutation-based test by shuffling the case-control status and re-selected top 20,000 CpGs based on the permuted data to construct module and test for association with CHD. In the permutation test, we found no inflated false positives due to the selection of top 20,000 CpGs (the most significant module has $P > 0.032$, *Figure 2—figure supplement 1*).

For CHD-associated modules ($P < 0.05$/the number of modules, Bonferroni correction), we performed gene enrichment analysis using the list of annotated genes from this module (DAVID, https://david.ncifcrf.gov/) (*Huang et al., 2007*), and further determined the significant CHD related CpGs within the module (module-specific FDR < 0.01). For CHD-associated loci, we further fitted logistic regression adjusting for the same set of covariates and all SVs to interpret the effect size better.

### Association between CHD-associated CpGs and aardiovascular risk factors

We investigated the associations between lifestyle factors and CHD-associated CpGs, with the methylation value as the outcome. Lifestyle factors included smoking, alcohol consumption, physical activity, dietary habit, and BMI.

If the lifestyle-methylation association suggests marginal significance ($P < 0.05$), we performed causal mediation analysis using parametric regression models, achieved by paramed package in STATA

(RRID:SCR_012763). Two models were estimated for each CpG: (1) a model for the mediator (methylation level as a continuous variable) conditional on exposure (the corresponding lifestyle factor) and covariates (age, sex, study area, fasting time, education level, marital status, the other four lifestyle factors, and batch); (2) a model for the risk of CHD conditional on exposure, the mediator, and covariates. We allowed for the presence of exposure-mediator interactions in the outcome regression model.

We aimed to calculate how much of the CHD risk associated with lifestyle factors (total effect, TE) was attributable to mediating effect of methylation level at a specific locus (natural indirect effect, NIE). The proportion attributable to the NIE was calculated as NIE divided by TE on log odds scale, with 0 indicating no mediation effect.

We also investigated the association between CHD-associated CpGs and cardiometabolic traits, with the cardiometabolic traits as the outcome. Cardiometabolic traits included systolic blood pressure (SBP), diastolic blood pressure (DBP), blood lipid level (TC, LDL-C, HDL-C, and TG), and random glucose.

For CpGs which were statistically significantly associated with any of the cardiometabolic traits, we calculated the mediation effect of methylation level on CHD through a specific cardiometabolic trait. Two models were estimated for each CpG: (1) a model for the mediator (the corresponding cardiometabolic risk factor) conditional on exposure (methylation level) and covariates (age, sex, study area, fasting time, education level, marital status, five lifestyle factors, and batch); (2) a model for the risk of CHD conditional on exposure, the mediator, and covariates.

In the analysis of blood pressure, we added 15 and 10 mmHg to the measured SBP and DBP respectively among participants who reported usage of blood pressure-lowering medications. In the analysis of random glucose, we additionally adjusted for treatment of diabetes at baseline.

We adjusted batch IDs instead of SVs in CpGs-CHD risk factor association analysis and the corresponding mediation analysis because SVs adjustment was more appropriate when the methylation value was treated as the outcome.

## Results

Baseline DNA methylation was measured for 494 CHD cases and 494 matched controls. After quality control, we included 491 cases free of

**Table 1.** Age-, sex- and study area-adjusted baseline characteristics of 982 participants according to the case or control status.

| Baseline characteristics | Cases (n = 491) | Controls (n = 491) | P value |
|---|---|---|---|
| Age, year | 50.6 | 49.5 | - |
| Female, % | 43.6 | 43.6 | - |
| Urban area, % | 20.6 | 20.6 | - |
| Middle school and above, % | 43.4 | 45.6 | 0.730 |
| Married, % | 90.4 | 94.7 | 0.028 |
| Family history of heart attack, % | 6.9 | 4.7 | 0.127 |
| Fasting time, h | 4.0 | 4.0 | - |
| Lifestyle factors | | | |
| Daily tobacco smoker, % | 46.6 | 40.3 | 0.004 |
| Daily alcohol drinker, % | 9.0 | 10.0 | 0.455 |
| Physical activity, MET-h/day | 22.0 | 23.9 | 0.097 |
| Diet score | 2.3 | 2.5 | 0.001 |
| Vegetables 7 days/week, % | 92.7 | 91.0 | 0.278 |
| Fruit 7 days/week, % | 9.4 | 13.8 | 0.030 |
| Read meat <7 days/week, % | 79.2 | 80.0 | 0.600 |
| Soybean product ≥4 days/week, % | 5.9 | 9.6 | 0.026 |
| Fish ≥1 days/week, % | 24.6 | 28.9 | 0.022 |
| Coarse grains ≥ 4 days/week, % | 22.8 | 24.6 | 0.047 |
| Body mass index, kg/m² | 23.9 | 23.3 | 0.002 |
| Metabolic risk factors | | | |
| Prevalent hypertension, % | 52.5 | 29.9 | < 0.001 |
| Prevalent diabetes, % | 10.0 | 4.5 | 0.004 |
| Blood lipids | | | |
| Total cholesterol, mmol/L | 4.69 | 4.52 | 0.005 |
| LDL-C, mmol/L | 2.35 | 2.21 | 0.003 |
| HDL-C, mmol/L | 1.22 | 1.18 | 0.025 |
| Triglyceride, mmol/L | 2.20 | 2.01 | 0.064 |

The results are presented as means or percentages. P values were not showed for matched factors. MET = metabolic equivalent of task; LDL-C = low-density lipoprotein cholesterol; HDL-C = high-density lipoprotein cholesterol.

CHD at baseline and developing CHD during follow up and 491 controls free of CHD at baseline and follow up and matched for birth year, age at baseline, sex, study area, and area, hours fasting prior to blood draw.

The mean age was 50.6 ± 7.6 years for incident CHD cases and 49.5 ± 7.3 years for matched controls. Compared with control participants, the CHD cases were more likely to be daily smokers, have unhealthy dietary habits, and have higher BMI. CHD cases also had a higher prevalence of hypertension and diabetes and worse lipid profile at baseline (*Table 1*).

## Association between single DNA methylation marker and incident CHD

EWAS revealed an excess of association across a range of P thresholds (*Supplementary file 2A*). The genomic inflation factor was 1.09 after adjustment for covariates and surrogate variables (SV). The Quantile-Quantile plot (Q-Q plot) indicated little residual confounding (*Supplementary file 2B*). Methylation markers at two genetic regions were associated with incident CHD at FDR < 0.05 (*Table 2* and *Supplementary file 2B*). The corresponding p-value of the FDR = 0.05 threshold was 2.01E-07. The adjusted difference (standard error, SE) in methylation level between cases and controls was –0.003 (0.0006) for cg23398826 (*P* = 1.57E-08), which was annotated to *SNX30*. The SD of the beta value of cg23398826 was 0.008. The odds ratio (OR) (95 % confidence interval [CI]) for incident CHD was 0.56 (0.45, 0.70) per SD increase in methylation level at cg23398826. The corresponding adjusted difference (SE) for cg02386575 was 0.006 (0.0011; *P* = 9.61E-08), annotated to *IMPDH2* and *QRICH1* (*Table 2*). The SD was 0.016. The OR (95% CI) for per SD increase in cg02386575 was 2.00 (1.57, 2.56).

## Association between weighted gene co-methylation network and CHD risk

We used weighted gene co-methylation network analysis (WGCNA) (*Langfelder and Horvath, 2008*) to identify potential co-methylation network related to CHD. This method can be used for identifying clusters of highly correlated co-methylation genes and relating modules to external sample traits to find biologically or clinically significant modules. Two samples were outliers and excluded during the sample clustering step. We included 491 cases and 489 controls in the following analysis. A total of five modules were produced in the clustering step of WGCNA (*Figure 2*). One module (called: Brown module), containing 2,106 CpG sites, was associated with incident CHD after adjustment for covariates and all SVs (*P* < 0.05/the number of modules, *P* = 6.41E-08).

Gene enrichment analysis of the annotated genes of 2106 CpG sites in this module revealed six annotation clusters with at least one term having an FDR < 0.05 (*Supplementary file 2C*). These annotation clusters were significantly enriched in terms associated with intracellular signaling (zinc-finger, pleckstrin homology domains, C2 domains, and protein kinase activity) and transcription regulation. Annotated genes in this module were also enriched in genes associated with tobacco use disorder, stroke, and kidney disease (the Genetic Association Database). We performed the visualization of the Brown module (*Hu et al., 2008*) and found CpGs annotated to *ZNF790*, *CC2D1B*, *TBR1*, *RERE*, and *PLXNB2* had the most connections with other genes (*Figure 2—figure supplement 2*).

Within the Brown module, 24 CpGs were significantly associated with CHD (module-specific FDR < 0.01), with P ranging from 1.10E-04–9.61E-08 (*Table 2*). Together with two CpGs identified from single-maker tests, a total of 25 CpGs were associated with CHD, with OR (95 % confidence interval) ranging from 0.53 (0.38, 0.73) for cg26334131 to 2.18 (1.47, 3.23) for cg21210537 (*Table 2*).

## CHD-associated CpGs and cardiovascular risk factors

Methylation level at cg08106661 was associated with the average number of cigarette equivalents per day (effect size = 1.50E-04, SE = 4.67E-05, *P* = 0.001; *Table 3*). Further mediation analysis revealed that 28.5 % of the smoking-associated CHD risk was mediated through methylation level at cg08106661 (*P* = 0.036). We also found three loci associated with diet score and two loci associated with BMI, but no statistically significant mediation effect was noted (*P* > 0.05). Alcohol consumption and physical activity were not associated with any of the CHD-associated CpGs (*Table 3* and *Table 3—source data 1*).

Compared with participants in the bottom quartile of methylation level at cg23398826, those in the top quartile had 6.4 (SE 2.1) mmHg lower SBP (*P* = 0.003) and 3.6 (1.2) mmHg lower DBP (*P* = 0.003). The proportions of reduced CHD risk associated with cg23398826 mediated by SBP and DBP were 7.65 % (*P* = 0.003) and 6.39 % (*P* = 0.006), respectively (*Table 4*, *Table 4—source data 1* and *Table*

**Table 2.** Associations of 25 significant CpGs with the risk of coronary heart disease.

| Chr | Position (hg19) | CpG | SD | Gene | Relation to gene | EWAS β‡ | P | FDR | WGCNA* Module-specific FDR | Odds Ratio† (95% CI) |
|---|---|---|---|---|---|---|---|---|---|---|
| 9 | 115513036 | cg23398826 | 0.008 | SNX30 | TSS200 | –0.003 | 1.57E-08 | 0.012 | 1.05E-04 | 0.56 (0.45, 0.70) |
| 3 | 49068057 | cg02386575 | 0.016 | IMPDH2 | TSS1500 | 0.006 | 9.61E-08 | 0.036 | 2.02E-04 | 2.00 (1.57, 2.56) |
|  |  |  |  | QRICH1 | Body |  |  |  |  |  |
| 19 | 37329330 | cg10400937 | 0.007 | ZNF790 | TSS200 | 0.002 | 1.09E-05 | 0.288 | 0.009 | 1.53 (1.24, 1.89) |
| 12 | 131758671 | cg20562821 | 0.022 | (RPS6P20 §) |  | 0.005 | 2.42E-05 | 0.288 | 0.009 | 1.72 (1.28, 2.31) |
| 6 | 34855635 | cg08106661 | 0.016 | TAF11 | 1stExon | 0.003 | 3.16E-05 | 0.305 | 0.009 | 1.87 (1.35, 2.59) |
|  |  |  |  | ANKS1A | TSS1500 |  |  |  |  |  |
| 1 | 153203211 | cg11630610 | 0.019 | (MIR584§) |  | 0.005 | 3.83E-05 | 0.329 | 0.009 | 1.77 (1.36, 2.32) |
| 1 | 8426319 | cg20302171 | 0.018 | RERE | 5'UTR | –0.004 | 4.29E-05 | 0.340 | 0.009 | 0.55 (0.42, 0.73) |
| 11 | 63909324 | cg26334131 | 0.025 | MACROD1 | Body | –0.005 | 4.44E-05 | 0.340 | 0.009 | 0.53 (0.38, 0.73) |
| 20 | 2444631 | cg07560408 | 0.018 | SNORD119 | TSS1500 | –0.005 | 4.46E-05 | 0.340 | 0.009 | 0.60 (0.47, 0.77) |
|  |  |  |  | SNRPB | Body |  |  |  |  |  |
| 19 | 46522185 | cg21210537 | 0.027 | MIR769 | TSS200 | 0.004 | 4.85E-05 | 0.356 | 0.009 | 2.18 (1.47, 3.23) |
| 20 | 60546782 | cg15833447 | 0.021 | (TAF4§) |  | 0.006 | 5.55E-05 | 0.375 | 0.009 | 1.50 (1.20, 1.88) |
| 11 | 94963255 | cg02591826 | 0.005 | LOC100129203 | TSS200 | 0.002 | 5.70E-05 | 0.375 | 0.009 | 1.52 (1.23, 1.87) |
| 7 | 100861083 | cg16639138 | 0.006 | ZNHIT1 | 5'UTR/1stExon | 0.002 | 6.46E-05 | 0.375 | 0.009 | 1.52 (1.24, 1.86) |
|  |  |  |  | PLOD3 | TSS200 |  |  |  |  |  |
| 6 | 27863042 | cg01545454 | 0.007 | (HIST1H2BO§) |  | 0.002 | 7.29E-05 | 0.378 | 0.009 | 1.64 (1.26, 2.13) |
| 1 | 203242409 | cg07219103 | 0.008 | (CHIT1§) |  | 0.002 | 7.35E-05 | 0.378 | 0.009 | 1.78 (1.28, 2.47) |
| 22 | 23994996 | cg05681643 | 0.018 | GUSBP11 | Body | 0.004 | 7.42E-05 | 0.378 | 0.009 | 1.60 (1.24, 2.08) |
| 2 | 88991375 | cg06358566 | 0.009 | RPIA | 1stExon | –0.002 | 7.74E-05 | 0.385 | 0.009 | 0.62 (0.48, 0.80) |
| 2 | 162273185 | cg19583211 | 0.016 | TBR1 | 1stExon | –0.003 | 7.97E-05 | 0.385 | 0.009 | 0.56 (0.41, 0.77) |
| 20 | 3613189 | cg10643850 | 0.025 | ATRN | Body | 0.004 | 8.04E-05 | 0.385 | 0.009 | 1.97 (1.37, 2.82) |
| 17 | 17460905 | cg13311494 | 0.016 | PEMT | Body | –0.005 | 8.50E-05 | 0.397 | 0.009 | 0.64 (0.52, 0.79) |
| 1 | 179852195 | cg11754670 | 0.009 | TOR1AIP1 | Body | 0.001 | 8.84E-05 | 0.398 | 0.009 | 2.04 (1.40, 2.97) |
| 15 | 74928935 | cg05740632 | 0.014 | EDC3 | Body | –0.004 | 9.07E-05 | 0.398 | 0.009 | 0.62 (0.49, 0.78) |
| 11 | 1972510 | cg08484100 | 0.023 | MRPL23 | Body | –0.004 | 9.19E-05 | 0.398 | 0.009 | 0.54 (0.40, 0.74) |
| 1 | 52822428 | cg24792179 | 0.019 | CC2D1B | Body | 0.004 | 9.87E-05 | 0.410 | 0.009 | 1.79 (1.36, 2.35) |
| 7 | 68973036 | cg22794712 | 0.021 | (LOC105507468§) |  | –0.006 | 1.10E-04 | 0.413 | 0.010 | 0.63 (0.50, 0.80) |

*cg23398826 in the Turquoise module, all other CpGs in the Brown module.

†Odds ratios were for per standard deviation increase in DNA methylation level.

‡Effect sizes were calculated based on normalized methylation values, denoting the methylation difference between cases and controls.

§For inter-genic CpG sites, R package FDb.InfiniumMethylation.hg19 was used to locate the nearest annotated gene.

CpG = cytosine-phosphoguanine site. Chr = chromosome. EWAS = epigenome wide association. WGCNA = weighted gene co-methylation network analysis. FDR = false discovery rate. CI = confidence interval. TSS200 = within 200 bp from transcription start site. TSS1500 = within 1500 bp from transcription start site. Body = the CpG is in gene body. 1stExon = the first exon. and UTR = untranslated region.

*4—source data 2*). The analysis also showed statistically significant mediation of methylation level at cg13311494 (annotated to *PEMT*) on CHD risk through SBP and DBP, with the mediation proportions of 15.61% and 12.38%, respectively. Four TC-related (*Table 4* and *Table 4—source data 3*), six LDL-C

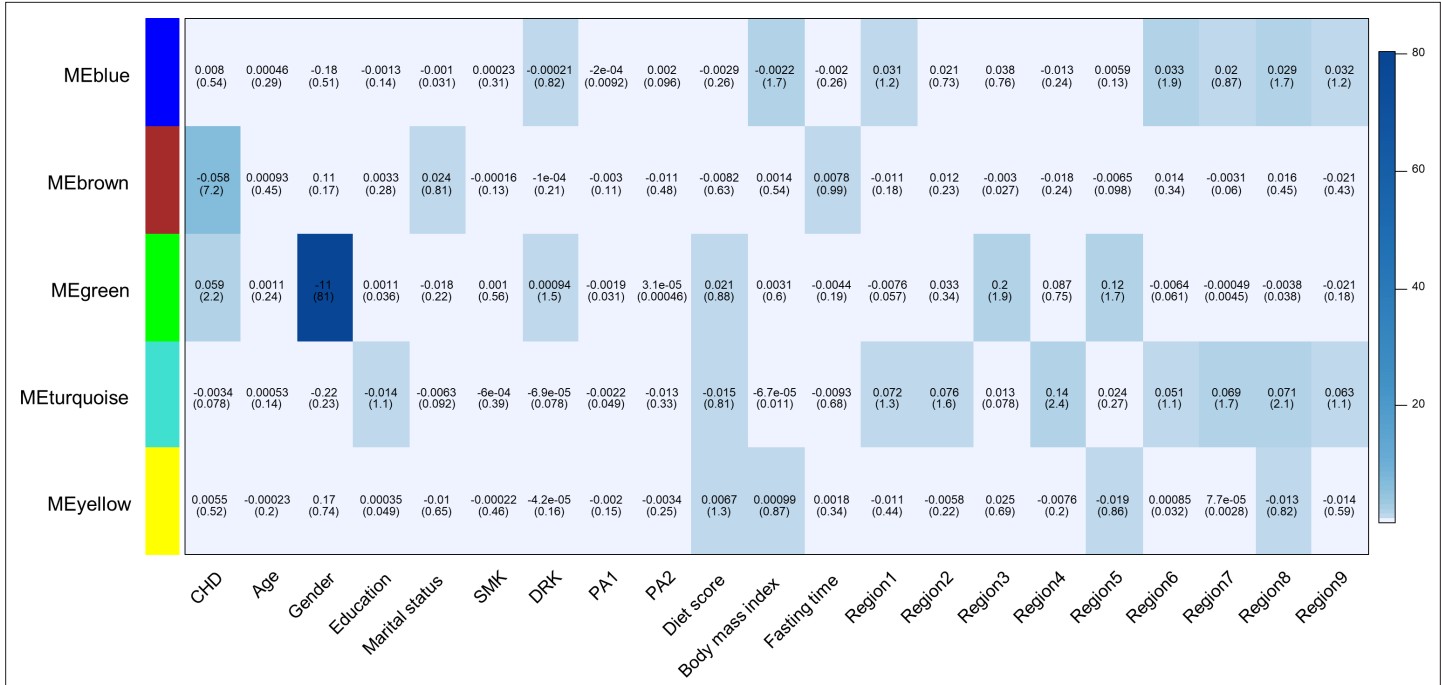

**Figure 2.** Heatmap of association with methylation network modules. Correlation coefficient and -log10(P) (inside the bracket) were reported; the degree of -log10(p) is illustrated with the color legend. Linear regressions were fitted with inverse normal transformed module eigengene (ME) as dependent variables; coronary heart disease (CHD1) as indicator; and age, sex, education, marital status, smoking (SMK), drinking (DRK), physical activity (PA1 and PA2 as the second and third tertile respectively), diet score, body mass index, fasting time, study area and all surrogate variables as covariates.

The online version of this article includes the following figure supplement(s) for figure 2:

**Figure supplement 1.** Permutation test to confirm the validity of weighted gene co-methylation network analysis.

**Figure supplement 2.** Visualization of the brown module.

related (*Table 4* and *Table 4—source data 4*), two HDL-C related (*Table 4* and *Table 4—source data 5*), and six random glucose-related (*Table 4* and *Table 4—source data 6*) CpGs had P for trend <0.05. However, no statistically significant mediation effect was shown for the associations between corresponding methylation level on CHD through these traits. TG was not associated with any of the CHD-associated CpGs (*Table 4—source data 7*).

To test the robustness of the findings, we restricted both CpGs-CHD and CpGs-SBP/DBP analyses to participants without the usage of blood pressure-lowering drugs at baseline (n = 880). The association magnitudes of methylation level with CHD were mostly unchanged (*Supplementary file 2D*). Such restriction slightly attenuated the association of methylation level at cg23398826 and cg13311494 with SBP and DBP (*Supplementary file 2E and F*). In the association analyses of 25 CHD-associated CpGs and cardiovascular risk factors, we also performed smart SVA for each trait. Adjustment for all SVs instead of batch did not change the association materially (*Table 4—source data 1*; *Table 4—source data 2*; *Table 4—source data 3*; *Table 4—source data 4*; *Table 4—source data 5*; *Table 4—source data 6*; *Table 4—source data 7*).

## Discussion

In this prospective study of middle-aged Chinese, we found methylation at 25 CpGs across the genome were associated with incident CHD risk over the next 10 years. One SD increase in methylation level of identified CpGs was associated with differences in CHD risk, ranging from a 47 % decrease (cg26334131) to a 118 % increase (cg21210537) in CHD risk. Further mediation analyses revealed two potential pathways to CHD risk, one with methylation at cg08106661 mediating the impact of smoking, and the other with blood pressure mediating the impact of methylation at cg23398826 and cg13311494. One co-methylation network suggested a role for intracellular signaling in CHD risk.

**Table 3.** Associations between lifestyle factors and methylation level of identified CpGs, and the risk of coronary heart disease mediated through methylation level of CpG sites.

| | Effect size (SE) | P | Mediation effect | |
| | | | Proportion mediated, % | P |
| --- | --- | --- | --- | --- |
| **Smoking, no. of cigarettes/day** | | | | |
| cg08106661 | 1.50E-04 (4.67E-05) | 0.001 | 28.50 | 0.036 |
| **Diet score (ranging 0-6)** | | | | |
| cg21210537 | 3.60E-03 (1.27E-03) | 0.005 | 4.66 | 0.206 |
| cg10643850 | 2.57E-03 (1.26E-03) | 0.042 | -6.91 | 0.088 |
| cg05740632 | 1.37E-03 (6.88E-04) | 0.047 | 11.30 | 0.068 |
| **Body mass index, kg/m²** | | | | |
| cg20302171 | 3.90E-04 (1.67E-04) | 0.020 | -2.87 | 0.267 |
| cg08484100 | 4.17E-04 (2.10E-04) | 0.048 | -1.91 | 0.373 |

Linear regression was fitted by including all five lifestyle factors (smoking, alcohol consumption, physical activity, diet score, and body mass index) simultaneously in the same model, with methylation values as dependent variables, and age, sex, study area, fasting time, education level, marital status and batch as covariates. CpG = cytosine-phosphoguanine site; SE = standard error. Alcohol consumption and physical activity were not associated with any of the coronary heart disease-associated CpGs. Details were reported in the Table 3—source data 1.

The online version of this article includes the following source data for table 3:

**Source data 1.** Association between lifestyle factors and identified CpGs.

We summarized the annotated or nearest annotated gene of the identified CHD-associated CpGs in our study and the previous GWAS findings (*Supplementary file 2G*). Four of the total 25 identified CpGs map to genes that have been reported in association with cardiovascular disease in previous GWAS studies. CpG cg08106661 maps to the *ANKS1A* (Ankyrin repeat and SAM domain-containing protein 1 A) gene with critical roles in regulating the epidermal growth factor receptor (EGFR). Activation of EGFR has been implicated in endothelial dysfunction, atherogenesis, and cardiac remodeling (*Makki et al., 2013*). SNPs in *ANKS1A* have been consistently linked to CHD and smoking behaviour in different populations (*Dichgans et al., 2014*; *Charmet et al., 2018*; *Schunkert et al., 2011*). Furthermore, our mediation analysis noted that more than 25 % of the increased CHD risk related to smoking was mediated through methylation level at cg08106661. Our results provide evidence that smoking-induced epigenetic modification of DNA may play an important part in the underlying pathway from smoking to CHD.

Five identified CHD-associated CpG loci were linked to blood pressure in previous GWAS studies. CpG cg23398826 was located within 200 bp from the transcription start site of the *SNX30* (Sorting Nexin 30) gene, a member of the sorting nexin family which plays a vital role in endocytic trafficking. The perturbation of this process may lead to impaired homeostatic responses and possibly disease states, including cardiovascular disease (*Yang et al., 1979*). *SNX30* has been reported in a GWAS of DBP night-to-day ratio (*Rimpelä et al., 2018*). The methylation level at CpG cg23398826 was found to be associated with SBP and DBP in our study. Further mediation analysis showed that blood pressure mediated ~10 % of the reduced CHD risk related to methylation at cg23398826, suggesting that such epigenetic regulation might exert an important influence on blood pressure and subsequent risk of CHD. However, methylation level and blood pressure were both measured at baseline. We note that directional association between methylation and blood pressure is still unknown.

WGCNA identified one CHD-associated gene co-methylation network. Gene members of this network were enriched in several protein domains, molecule function, and pathways that are involved in intracellular signaling. Cells can respond to the environment and extracellular cues by this vital mechanism (*Schulman, 2013*). One previous study using an in vitro model of cardiac hypertrophy

**Table 4.** Associations between quartile methylation level of identified CpGs and cardiometabolic traits, and the risk of coronary heart disease mediated through different cardiometabolic traits.

| | Quartile 1 vs. 4 | | | Mediation effect | |
| | | | | | P |
| | Effect size (SE)[*] | P | P for trend | Proportion mediated, % | |
|---|---|---|---|---|---|
| Systolic blood pressure[†], mmHg | | | | | |
| cg23398826 | -6.410 (2.118) | 0.003 | <0.001 | 7.65 | 0.003 |
| cg13311494 | -6.580 (2.122) | 0.002 | 0.020 | 15.61 | 0.031 |
| Diastolic blood pressure[†], mmHg | | | | | |
| cg23398826 | -3.574 (1.218) | 0.003 | <0.001 | 6.39 | 0.006 |
| cg13311494 | -3.650 (1.221) | 0.003 | 0.029 | 12.38 | 0.045 |
| Total cholesterol, mmol/L | | | | | |
| cg26334131 | 0.197 (0.089) | 0.026 | 0.003 | -31.62 | 0.168 |
| cg05740632 | 0.163 (0.089) | 0.066 | 0.013 | -3.21 | 0.126 |
| cg21210537 | 0.175 (0.094) | 0.064 | 0.027 | -8.19 | 0.197 |
| cg19583211 | -0.064 (0.088) | 0.466 | 0.047 | 2.73 | 0.270 |
| Cholesterol in LDL, mmol/L | | | | | |
| cg26334131 | 0.110 (0.063) | 0.079 | 0.007 | -32.14 | 0.135 |
| cg20302171 | 0.107 (0.063) | 0.09 | 0.029 | -10.48 | 0.161 |
| cg05740632 | 0.109 (0.063) | 0.083 | 0.019 | -3.38 | 0.110 |
| cg19583211 | -0.078 (0.062) | 0.208 | 0.020 | 3.60 | 0.210 |
| cg13311494 | -0.117 (0.062) | 0.06 | 0.027 | 3.70 | 0.208 |
| cg21210537 | 0.126 (0.067) | 0.059 | 0.044 | -8.55 | 0.177 |
| Cholesterol in HDL, mmol/L | | | | | |
| cg15833447 | 0.037 (0.026) | 0.154 | 0.013 | -10.77 | 0.180 |
| cg21210537 | 0.040 (0.027) | 0.146 | 0.019 | 7.30 | 0.235 |
| Random blood glucose[‡], mmol/L | | | | | |
| cg10400937 | 0.551 (0.231) | 0.017 | 0.003 | 6.72 | 0.107 |
| cg01545454 | 0.203 (0.234) | 0.385 | 0.006 | 9.58 | 0.097 |
| cg11754670 | 0.466 (0.236) | 0.049 | 0.005 | 36.39 | 0.086 |
| cg26334131 | -0.517 (0.234) | 0.028 | 0.018 | 32.70 | 0.109 |
| cg07219103 | 0.578 (0.244) | 0.018 | 0.032 | 6.17 | 0.135 |
| cg20302171 | -0.556 (0.234) | 0.018 | 0.027 | 15.77 | 0.123 |

[*]Effect sizes denoted the differences of metabolic traits between the top and bottom quartile methylation level. Details of other quartiles were reported in the Table 4—source data 1.

[†]We added 15 and 10 mmHg to the measured systolic blood pressure and diastolic blood pressure respectively among participants who reported usage of blood pressure-lowering medications.

[‡]Additionally adjusted for treatment of diabetes (yes or no) at baseline. Multivariable model was adjusted for: age, sex, education level, marital status, smoking, drinking, physical activity, dietary score, body mass index, fasting time, study area, and batch. The CpGs which were significantly associated with any metabolic risk factors were reported. Details of other CpGs were reported in the Supplementary tables. CpG = cytosine-phosphoguanine site; SE = standard error; LDL = low-density lipoprotein; HDL = high-density lipoprotein.

The online version of this article includes the following source data for table 4:

**Source data 1.** Association between quartile methylation level of identified CpGs and systolic blood pressure (mmHg).

*Table 4 continued on next page*

**Table 4 continued**
**Source data 2.** Association between quartile methylation level of identified CpGs and diastolic blood pressure (mmHg).
**Source data 3.** Association between quartile methylation level of identified CpGs and total cholesterol (mmol/L).
**Source data 4.** Association between quartile methylation level of identified CpGs and cholesterol in low-density lipoprotein (mmol/L).
**Source data 5.** Association between quartile methylation level of identified CpGs and cholesterol in high-density lipoprotein (mmol/L).
**Source data 6.** Association between quartile methylation level of identified CpGs and random glucose (mmol/L).
**Source data 7.** Association between quartile methylation level of identified CpGs and triglyceride (mmol/L).

revealed that differentially methylated promoters were involved in the intracellular signaling process (*Stenzig et al., 2015*). Nevertheless, these findings could only be interpreted as a possible functional indication and may stimulate future studies in translating these findings toward a better understanding of disease mechanisms.

Previous epigenome-wide studies of CHD in Asian population were all cross-sectional design with relatively small sample size (*Nakatochi et al., 2017*; *Sharma et al., 2014*; *Li et al., 2017*), in which the changes in DNA methylation at identified CpGs might be a result of disease state. Only two studies have employed prospective design (*Guarrera et al., 2015*; *Golareh et al., 2019*). One of them used a meta-analysis of nine population-based cohorts from the US and Europe (11,461 participants, mean age 64 years, mean follow-up time 11.2 years) to analyze CHD-associated DNA methylation at a single-nucleotide resolution (*Golareh et al., 2019*). Based on HumanMethylation450 BeadChip data, methylation levels at 52 CpG sites were identified to be associated with incident CHD or myocardial infarction. Differences in the nature of study design, genetic background and age distribution of the study population, follow-up time, and coverage of CpG sites might explain that our findings did not overlap with the previous CHD EWAS in either Asian or European population. Three identified loci in the present study could be replicated at the gene level (*ANKS1A*, *RERE*, and *EDC3*), by a small study which investigated differential DNA methylation loci using 15 donor-matched healthy and atherosclerotic human aorta samples in Spain (*Silvio et al., 2014*). Similar DNA methylation patterns at certain genes might be consistent in blood leukocytes and atherosclerotic lesions during the development of CHD.

Our study is by far the first prospective and the largest EWAS of CHD in the Asian population. The prospective design allowed us to identify loci where changes in DNA methylation potentially predict the risk of future CHD. The use of the latest DNA methylation array that covers over 850,000 CpG methylation sites provides extensive coverage of CpG islands, genes, and enhancer, however, also increases the burden of multiple hypothesis testing. The causal mediation analysis was added to help understand the functional potential of identified loci.

Our study has limitations. Although we have made a comprehensive adjustment for preselected potential confounders and also used the recommended SVA method to remove the unknown confounding effect, residual confounding is still possible. However, the potential inflation due to unadjusted confounding effect was small as indicated by the Q-Q plot. The cases and controls were not randomized on arrays. Adjusting batch effect may lose power to some extent. Future studies with samples at multiple time points preceding CHD onset are expected to provide insights into the role of dynamic changes of methylation and expression level in the progress of CHD.

We presented novel findings on associations of leukocyte DNA methylation at 25 CpGs with CHD risk over the next ten years among Chinese. Our findings also suggested the possible role of epigenetic regulations in the pathways to CHD risk, through or from lifestyle and cardiometabolic factors. Studies are warranted to validate our findings, elucidate the functional mechanisms of newly identified CpGs, and further translate our findings toward preventive or clinical implications.

## Acknowledgements

The most important acknowledgement is to the participants in the study and the members of the survey teams in each of the 10 regional centres, as well as to the project development and management teams based at Beijing, Oxford and the 10 regional centres.

# Additional information

## Competing interests
China Kadoorie Biobank Collaborative Group: The members of the steering committee and collaborative group are listed in the Supplementary file 1.. The other authors declare that no competing interests exist.

## Funding

| Funder | Grant reference number | Author |
|---|---|---|
| National Natural Science Foundation of China | 81390544 | Jun Lv |
| National Natural Science Foundation of China | 91846303 | Jun Lv |
| Wellcome Trust | 202922/Z/16/Z | Zhengming Chen |
| National Key Research and Development Program of China | 2016YFC0900500 | Yu Guo |

The funders had no role in study design, data collection and interpretation, or the decision to submit the work for publication.

## Author contributions
Jiahui Si, Conceptualization, Formal analysis, Validation, Visualization, Writing – original draft, Writing – review and editing; Songchun Yang, Formal analysis, Validation, Writing – review and editing; Dianjianyi Sun, Wei Chen, Methodology, Software, Writing – review and editing; Canqing Yu, Iona Y Millwood, Robin G Walters, Ling Yang, Yiping Chen, Huaidong Du, Data curation, Investigation, Writing – review and editing; Yu Guo, Data curation, Funding acquisition, Investigation, Writing – review and editing; Yifei Lin, Methodology, Writing – review and editing; Yujie Hua, Jingchao Liu, Investigation, Writing – review and editing; Junshi Chen, Data curation, Project administration, Writing – review and editing; Zhengming Chen, Data curation, Funding acquisition, Project administration, Writing – review and editing; Jun Lv, Conceptualization, Data curation, Funding acquisition, Investigation, Methodology, Project administration, Resources, Supervision, Writing – review and editing; Liming Liang, Conceptualization, Methodology, Resources, Software, Supervision, Writing – review and editing; Liming Li, Conceptualization, Data curation, Funding acquisition, Investigation, Project administration, Resources, Supervision, Writing – review and editing

## Author ORCIDs
Jiahui Si (iD) http://orcid.org/0000-0003-0827-4973
Canqing Yu (iD) http://orcid.org/0000-0002-0019-0014
Jun Lv (iD) http://orcid.org/0000-0001-7916-3870

## Ethics
The study protocol was approved by the Ethics Review Committee of the Chinese Center for Disease Control and Prevention (Beijing, China), the Oxford Tropical Research Ethics Committee, University of Oxford (UK), and Peking University Institutional Review Board (Beijing, China). All participants provided written informed consent.

## Decision letter and Author response
Decision letter https://doi.org/10.7554/eLife.68671.sa1
Author response https://doi.org/10.7554/eLife.68671.sa2

# Additional files

## Supplementary files
- Supplementary file 1. Members of the China Kadoorie Biobank collaborative group.
- Supplementary file 2. Detailed results and interpretation of the present study. (A) Number

of methylation markers associated with incident coronary heart disease across the range of P thresholds in the epigenome-wide association study. (B) Manhattan plot (A) and QQ plot (B) of the P values of the associations between each cytosine-phosphoguanine (CpG) site and incident coronary heart disease. In the Manhattan plot, the red line represents -log10(P) at false discovery rate (FDR) = 0.05. (C) Gene enrichment analysis of 2106 probes from the brown module which was significantly associated with coronary heart disease. (D) Associations of 25 significant CpGs with risk of coronary heart disease among 880 participants without usage of blood pressure lowering drug. (E) Association between quartile methylation level of identified CpGs and systolic blood pressure* (mmHg). (F) Association between quartile methylation level of identified CpGs and diastolic blood pressure* (mmHg). (G) The annotated or nearest annotated gene of the identified CHD-associated CpGs in our study and the previous GWAS finding.

- Transparent reporting form
- Source code 1. Single DNA methylation marker and incident CHD.
- Source code 2. Weighted gene co-methylation network and incident CHD.
- Source code 3. Mediation analysis.
- Source code 4. Baseline characteristics according to the case or controlstatus.
- Reporting standard 1. STROBE Statement—Checklist of items that should be included in reports of *case-control studies*.

### Data availability

According to the Regulation of the People's Republic of China on the Administration of Human Genetic Resources, we are not allowed to provide Chinese human clinical and genetic data abroad without an official approval. The process of obtaining official approval usually takes 2-3 months. According to our previous experience, we can make the raw data of part data (significant CpGs that were found in our study), not all data available after the approval. For researchers who are interested to access the original data, the access policy and procedures are available at https://www.ckbiobank.org/site/. In brief, the China Kadoorie Biobank (CKB) is being conducted jointly by the Peking University (PKU) in Beijing, the Clinical Trial Service Unit (CTSU), and Nuffield Department of Population Health, University of Oxford. Requesters should be employees of a recognised academic institution, health service organisation or charitable research organisation with experience in medical research. Requestors should be able to demonstrate, through their peer reviewed publications in the area of interest, their ability to carry out the proposed study. After registration, details of the required information are provided on the CKB Data Access System. The CKB Access Team will review and respond to data requests within 6-8 weeks. We are providing our syntax of statistical analysis and the source data for Table 3 and Table 4. Figure 2 is the direct output from R software.

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
