## [Decision Letter]

**Acceptance summary:**

The authors present the first large long term study in a non white population of the association between epigenetic changes and coronary heart disease risk over ten years. In this well conducted study there are novel findings for the increased risk of smoking and for impact on blood pressure. This advances our overall knowledge of epigenetic regulation in the pathways to coronary heart disease.

**Decision letter after peer review:**

Thank you for submitting your article "Epigenome-wide analysis of DNA methylation and coronary heart disease: a nested case-control study" for consideration by *eLife*. Your article has been reviewed by 3 peer reviewers, including Edward D Janus as Reviewing Editor and Reviewer #1, and the evaluation has been overseen by Matthias Barton as the Senior Editor. The following individual involved in review of your submission has agreed to reveal their identity: Ida Karlsson (Reviewer #3).

The overall comments from the three reviewers are reproduced below. The paper is excellent. We have provided a list of comments and request you consider these in providing your revision.

Suggested recommendations:

1) Starting with the main concern, the lack of replication is of course an issue (as in most EWASs). Could you compare the top hits from the largest previous EWAS of incident CHD, to see if the direction of effect is similar and if the significance is suggestive in your data? That would tell us a little bit more about the comparability.

2) It would be helpful if some of the methods (and sample description) were provided along with the results, to better follow along the findings (without having to go to the end to read the results first). I found Figure 1 helpful in understanding the results and suggest that it could be presented with the results instead of the methods.

3) Some minor clarifications and justifications I would find helpful as a reader:

a. I am not very familiar with the co-methylation network approach. Could you provide a brief overview of the method (preferably along with the results)? As the majority of the findings stem from this method, a justification of its robustness would be helpful.

b. I agree with what you say in the discussion, that it is not possible to say what direction mediation works, but it would be good with a sentence to explain the reason of the directions selected to study mediation.

c. The sample selection: I assume you aimed to capture severe cases by including only fatal IHD or nonfatal MI counted as CHD? Why were individuals with neoplasms or cerebrovascular disease excluded?

4) A couple of questions regarding the statistical analyses:

a. The analysis description for single CpG sites only says linear regression. Were the analyses not matched for case-control status (i.e. conditional linear regression), and if so why not? That should be the more powerful and robust approach.

b. In relation to that, why adjust for the matching variables?

c. I am slightly worried about overadjustment, especially in the mediation analyses as several of the lifestyle covariates are likely correlated (e.g. physical activity and BMI). Might including these adjustments in the mediation analyses mask an effect? And for the main analyses, did you also test a simpler model with only basic adjustments for comparison?

5) Please check the number of incident cases and controls so the numbers are consistent in the abstract, figure 1, introduction, results etc. The numbers as currently shown vary slightly from 489 to 494.

6) In the first paragraph of the Results section please expand as the readers won't all look at the methodology section. See also (2) above.

Suggest – 491 cases free of CHD at baseline and developing CHD during follow up and 491 controls free of CHD at baseline and follow up and matched for age, gender, region and timing of blood sampling…

7) Table 1. Please show statistical significance p value for prevalent hypertension and diabetes and for lipids.

8) line 122, can you be specific about items related "healthy lifestyle" and are they known risk factors for CHD?

9) line 126, in my opinion, the genomic inflation factor is not helpful here as it is not a good indicator for EWAS. It is because CpGs are more much correlated than SNPs, and the inflation factor is very much dependent on the trait.

10) Line 130: The difference "-0.003" between cases and controls is very small and hard to detect. Can you also show the SD of the two CpGs, or simply plot their distributions in cases and controls.

11) Line 144: Why do you use "probe" not "CpG" in this paragraph?

12) Line 156 change none to no.

13) Line 242: Did you compare the maximum follow-up time and age distribution in the other two prospective publications? I believe the choice of different follow-up time can be an important factor as we are not sure how long it takes CpGs to have effects on CHD.

14) Line 312: "lab staff were blinded" does it also mean the cases and controls were randomized on arrays/batches. This is important as we don't want to lose study power by adjusting batch effect.

15) Line 365: Remove the repeated sentence "A total of 56 SVs were generated."

16) Line 434: How were cellular compositions controlled when you could not use SV here?

17) Can you also report the corresponding p-value of the FDR 0.05 threshold?

18) If you used β-values of DNA methylation in the analysis, please state it.*Reviewer #1:*

The authors examine the association between genome wide epigenetic changes and coronary heart disease in an effort to further explore the residual risk not explained by known risk factors

The major strengths are that this is a large longitudinal study over 10 years in non whites that is Chinese. Earlier studies have used small samples, been primarily in Western countries, have focused on selective genomic regions rather than the whole genome and with a few exceptions have been cross sectional which precludes establishment of a temporal and potentially causative relationship.

No major weaknesses are apparent. There has been a comprehensive effort to exclude confounders. In limitations the possibility of some residual confounders is noted.

In this well conducted study there are novel findings for impact of DNA methylation at 25 cytosine phosphate-guanine (CpG) dinucleotides, predominantly novel, on CHD risk; for the increased risk of smoking (25% mediated by methylation at one specific site) and for impact on blood pressure and intracellular signaling as well.

The authors have achieved their aims and the results support their conclusions.

This study advances our overall knowledge of epigenetic regulation in the pathways to coronary heart disease*Reviewer #2:*

This study used a matched case-control design to study the prospective effect of DNA methylation on coronary heart disease. The study data included 491 cases and 491 controls which can provide enough statistical power. The authors used proper statistical methods to achieve their aims, and the results support their conclusions. This study will contribute to our better understanding of the effect of DNA methylation on coronary heart disease.

*Reviewer #3:*

The authors present a thorough and very well-written EWAS of incident CHD in 491 cases and 491 matched controls. The sample was drawn from the China Kadoorie Biobank, with 10-years of follow-up, and the data had rich baseline information in addition to DNA methylation measures. Single site association analyses identified two sites (FDR p-value <0.05), and a network approach 24 sites (module-specific FDR<0.01; one site overlapping with single site associations). They further conducted mediation analyses, testing a wide range of cardiometabolic and lifestyle factors, and found evidence of mediation effects at some of the sites in relation to smoking and blood pressure. A weakness is that which is common to most EWASs, namely the lack of replication. However, the issue also highlights the importance of further studies in the field. The prospective nature as well as the Asian population further strengthens the importance, as both aspects are largely lacking in the field.

The methodology is sound and support the conclusions of the paper. I especially appreciate the addition of mediation analyses, as it provides a better understanding of potential biological pathways.

---

## [Author Response]

Suggested recommendations:1) Starting with the main concern, the lack of replication is of course an issue (as in most EWASs). Could you compare the top hits from the largest previous EWAS of incident CHD, to see if the direction of effect is similar and if the significance is suggestive in your data? That would tell us a little bit more about the comparability.

The previous largest EWAS of incident CHD used a meta-analysis of nine population-based cohorts and 11,461 participants from the United States and European countries^1^. Based on HumanMethylation450 BeadChip data, methylation levels at 30 CpG sites were identified to be associated with incident CHD, and 30 were associated with incident myocardial infarction (MI). The direction of effect was not quite the same. The effect of 55.2% (CHD-associated) and 51.9% (MI-associated) CpGs showed the same directions in our study and in the previous study. (Author response table 1) showed the β coefficient and p-value of these 60 top hits in our study and in the previous study. The genetic background of the study population might be an important factor for this difference and lack of comparability. Also, as the reviewer mentioned in comment 14, age distribution might explain the difference in results. The mean age of the participants in the present study was 50.1 years, and that of the previous EWAS was 64 years.

**Author response table 1. sa2table1:** 

Outcome	CpG name	beta coefficient in ourstudy	p-value in our study	beta coefficient in the previous study	p-valuein theprevious study	Gene
CHD	cg22617878	-5.89E-04	3.75E-01	−0.3719	1.99E-08	ATP2B2
	cg13827209	8.39E-04	4.47E-01	0.268	3.76E-08	TGFBR1
	cg14185717	/	/	−0.2878	1.38E-07	BNC2
	cg10307345	-3.62E-03	1.88E-01	−0.1480	1.86E-07	PTPN5
	cg13822123	-6.12E-04	1.45E-01	0.4138	2.03E-07	PSME4
	cg23245316	1.04E-03	1.18E-01	−0.4674	2.17E-07	TSSC1
	cg24977276	2.83E-03	4.60E-02	−0.3256	2.54E-07	GTF2I
	cg24447788	1.17E-03	3.20E-01	−0.2679	4.33E-07	(PTBP1**)
	cg08422803	1.35E-03	3.47E-01	0.1994	7.52E-07	ITGB2
	cg01751802	-1.43E-03	5.85E-01	0.1473	9.35E-07	KANK2
	cg02449373	2.25E-05	9.77E-01	0.3715	9.98E-07	FUT1
	cg02683350	-1.19E-03	3.62E-01	−0.5062	1.55E-06	ADAMTS2
	cg05820312	2.89E-04	7.87E-01	0.5031	1.60E-06	TRAPPC9
	cg06639874	-8.25E-04	6.41E-01	−0.2506	1.83E-06	MLPH
	cg06582394	-2.56E-03	1.49E-01	0.1657	1.90E-06	CASR
	cg02155262	-4.62E-04	3.07E-01	0.477	1.97E-06	AGA
	cg12766383	2.26E-03	3.11E-02	−0.6194	1.98E-06	UBR4
	cg05892484	5.55E-05	9.60E-01	−0.5020	2.01E-06	MAD1L1
	cg03031868	1.46E-04	8.65E-01	0.3461	2.29E-06	ESD
	cg25497530	-5.63E-03	1.45E-03	−0.2225	2.62E-06	PTPRN2
	cg06596307	-2.06E-03	1.24E-01	−0.4198	2.99E-06	IGF1R
	cg10702366	-2.35E-03	1.47E-01	−0.1093	3.09E-06	FGGY
	cg26470101	1.24E-03	4.46E-01	0.3052	3.09E-06	(DLX2**)
	cg26042024	1.66E-03	4.16E-01	−0.3109	3.13E-06	ZFAT
	cg00466121	7.27E-04	4.26E-01	0.4646	3.16E-06	ZNHIT6
	cg04987302	-1.96E-03	1.85E-01	−0.3378	3.71E-06	(OTX2-AS1**)
	cg08853494	1.82E-04	7.27E-01	0.221	4.03E-06	RCHY1;THAP6
	cg26467725	1.34E-03	2.02E-01	−0.4225	4.22E-06	SLCO3A1
	cg06442192	9.07E-04	5.78E-01	−0.5241	4.89E-06	ZNF541
	cg00393373	1.11E-03	2.92E-01	−0.3156	4.91E-06	ZNF518B
MI	cg22871797	4.77E-04	6.85E-01	−0.599	5.29E-08	CYFIP1
	cg24977276	2.83E-03	4.60E-02	−0.366	9.97E-08	GTF2I
	cg18598861	1.43E-03	1.98E-01	−0.671	1.61E-07	IRF9
	cg09777776	7.95E-04	7.38E-01	0.287	2.25E-07	ZNF254
	cg20545941	-3.22E-04	6.69E-01	−0.885	2.47E-07	MPPED1
	cg19935845	9.61E-04	5.30E-01	−0.336	4.65E-07	TNXB
	cg24423782	-2.37E-03	1.33E-01	−0.398	5.37E-07	MIR182
	cg00393373	1.11E-03	2.92E-01	−0.401	7.68E-07	ZNF518B
	cg00466121	7.27E-04	4.26E-01	0.487	7.79E-07	ZNHIT6
	cg19227382	/	/	−0.504	8.12E-07	CDH23
	cg03467256	/	/	−0.408	8.33E-07	HPCAL1
	cg25196881	3.45E-04	7.46E-01	−0.269	1.05E-06	(THBS1**)
	cg02321112	9.68E-05	8.99E-01	0.39	1.08E-06	(MNX1-AS1**)
	cg00355799	2.79E-04	8.27E-01	−0.216	1.40E-06	(LOC339529**)
	cg17556588	-1.20E-03	3.90E-01	−0.154	1.45E-06	PRRG4
	cg04987302	-1.96E-03	1.85E-01	−0.428	1.50E-06	(OTX2-AS1**)
	cg07289306	-1.45E-03	3.24E-02	0.616	1.71E-06	(MIR138-1**)
	cg05892484	5.55E-05	9.60E-01	−0.551	1.84E-06	MAD1L1
	cg10702366	-2.35E-03	1.47E-01	−0.150	2.11E-06	FGGY
	cg22618720	/	/	−0.424	2.37E-06	(MIR5095**)
	cg14010194	3.46E-04	7.84E-01	−0.484	2.71E-06	GUCA1B
	cg13827209	8.39E-04	4.47E-01	0.285	2.71E-06	TGFBR1
	cg24318598	-1.75E-03	2.59E-01	−0.254	2.79E-06	ANO1
	cg07015775	1.01E-03	1.36E-01	0.479	3.13E-06	ZNHIT6
	cg21018156	2.00E-03	1.39E-01	−0.135	3.17E-06	(LINC01312**)
	cg07475527	-1.64E-03	2.46E-01	−0.225	3.77E-06	(RCAN3**)
	cg20000562	2.06E-03	8.15E-02	0.218	3.93E-06	SFTA3
	cg07436807	3.57E-04	7.88E-01	−0.779	4.10E-06	STAMBPL1; ACTA2
	cg14029912	-1.62E-03	1.39E-01	−0.367	4.29E-06	(BHLHE40**)
	cg22871797	4.77E-04	6.85E-01	−0.599	5.29E-08	CYFIP1

2) It would be helpful if some of the methods (and sample description) were provided along with the results, to better follow along the findings (without having to go to the end to read the results first). I found Figure 1 helpful in understanding the results and suggest that it could be presented with the results instead of the methods.

We thank the reviewer for the thoughtful comment. We have revised the manuscript to make it easy to follow (Line 81-88).

3) Some minor clarifications and justifications I would find helpful as a reader:

a. I am not very familiar with the co-methylation network approach. Could you provide a brief overview of the method (preferably along with the results)? As the majority of the findings stem from this method, a justification of its robustness would be helpful.

We used weighted gene co-methylation network analysis^2^ to identify potential co-methylation network related to CHD. This method can be used to identify clusters of highly correlated co-methylation genes and relate modules to external sample traits to find biologically or clinically significant modules. By calculating correlations among the methylation level of selected CpG sites, we constructed a gene co-methylation network. We then identified gene modules using hierarchical clustering. Next, we related gene modules to CHD outcome.

For computational reasons, we selected the top 20,000 CHD-associated CpGs from single-marker tests. This is about the maximum number of CpGs the WGCNA package can handle on our high-performance computing cluster. A previous study has been restricted 23,000 probe sets to 3600 probes to test the robustness. They found that the module detection results were generally similar^2^. We also additionally carried out a permutation-based test by shuffling the case-control status and re-selected the top 20,000 CpGs based on the permuted data to construct module and test for association with CHD. We found no inflated false positives due to the selection of top 20,000 CpGs (the most significant module has P>0.032, Figure 2—figure supplement 1).

We have added a brief overview of the co-methylation network approach to the “Result” section (Line 111-114).

b. I agree with what you say in the discussion, that it is not possible to say what direction mediation works, but it would be good with a sentence to explain the reason of the directions selected to study mediation.

We thank the reviewer for the thoughtful comment. DNA methylation is responsive to environmental stimuli and unhealthy lifestyles. This makes DNA methylation a potential biomarker of environmental-related and lifestyle-driven diseases of adulthood, for example, metabolic dysfunction. Unhealthy lifestyles, together with metabolic dysfunction, will further increase the risk of cardiovascular disease. We have added to address this comment (Line 59-65).

c. The sample selection: I assume you aimed to capture severe cases by including only fatal IHD or nonfatal MI counted as CHD? Why were individuals with neoplasms or cerebrovascular disease excluded?

Yes, we included fatal IHD and nonfatal acute MI to capture severe cases. Previous studies suggested that DNA methylation was also a potential biomarker for neoplasms^3^ or cerebrovascular disease.^4,5^ Thus, cases with both CHD and cerebrovascular or neoplasms could present a mixture of epigenetic changes. We excluded participants who reported at baseline or have developed neoplasms or cerebrovascular diseases during follow-up to better capture the DNA methylation change associated with incident CHD.

4) A couple of questions regarding the statistical analyses:

a. The analysis description for single CpG sites only says linear regression. Were the analyses not matched for case-control status (i.e. conditional linear regression), and if so why not? That should be the more powerful and robust approach.

We didn’t use conditional linear regression in the analysis. Instead, we followed the recommendation by Leek JT, et al.^6^ to use simple linear regression when yielding surrogate variable analysis for removing batch effects and other unwanted variations in high-throughput experiments. Previous studies^7–10^ that used matched case-control design also didn’t perform conditional analysis for the matched factors, although the authors employed different methods to remove batch effects and other unmeasured cofounding (SVA,^9^ adjustment for principal component,^7,10^ or adjustment for technical variables directly^8^).

b. In relation to that, why adjust for the matching variables?

We agree with the reviewer’s consideration for the matched design. Besides the reason we mentioned in the last comment, we noticed that cases were still slightly older than controls despite they were already matched by age (Table 1). Thus, we adjusted for matching factors in the model instead of matching for case-control status in the analysis, same as previous studies.^7–10^

c. I am slightly worried about overadjustment, especially in the mediation analyses as several of the lifestyle covariates are likely correlated (e.g. physical activity and BMI). Might including these adjustments in the mediation analyses mask an effect? And for the main analyses, did you also test a simpler model with only basic adjustments for comparison?

We re-calculated SVs and adjusted for age, sex, body mass index, smoking status, education level, study area, and all SVs for comparison with the previous largest EWAS of CHD. We called this a basic model and our original model as a full model. Please find the results of these two models in Author response table 2. Adjustment for additional lifestyle covariates did not change the association materially.

**Author response table 2. sa2table2:** 

CpG		Basic model		Full model
	β	P	β	P
cg23398826	-0.003	3.44E-08	-0.003	1.57E-08
cg02386575	0.005	2.76E-07	0.006	9.61E-08
cg10400937	0.002	1.28E-05	0.002	1.09E-05
cg20562821	0.005	2.95E-05	0.005	2.42E-05
cg08106661	0.003	6.75E-05	0.003	3.16E-05
cg11630610	0.005	4.22E-05	0.005	3.83E-05
cg20302171	-0.004	3.86E-05	-0.004	4.29E-05
cg26334131	-0.005	3.51E-05	-0.005	4.44E-05
cg07560408	-0.004	5.73E-05	-0.005	4.46E-05
cg21210537	0.0043	4.81E-05	0.004	4.85E-05
cg15833447	0.005	0.000107	0.006	5.55E-05
cg02591826	0.002	4.97E-05	0.002	5.70E-05
cg16639138	0.002	0.000122	0.002	6.46E-05
cg01545454	0.002	6.38E-05	0.002	7.29E-05
cg07219103	0.002	6.77E-05	0.002	7.35E-05
cg05681643	0.004	7.33E-05	0.004	7.42E-05
cg06358566	-0.002	8.02E-05	-0.002	7.74E-05
cg19583211	-0.003	6.46E-05	-0.003	7.97E-05
cg10643850	0.004	0.000104	0.004	8.04E-05
cg13311494	-0.004	8.04E-05	-0.005	8.50E-05
cg11754670	0.001	0.000101	0.001	8.84E-05
cg05740632	-0.004	4.51E-05	-0.004	9.07E-05
cg08484100	-0.005	3.69E-05	-0.004	9.19E-05
cg24792179	0.004	0.000115	0.004	9.87E-05
cg22794712	-0.005	0.000173	-0.006	1.10E-04

Similarly, we also fitted basic models in the mediation analysis by including age, sex, BMI (exclude when BMI was exposure), smoking status (exclude when smoking was exposure), education level, study area, and batch as covariates. The results were largely retained (See Author response table 3).

**Author response table 3. sa2table3:** 

	Basic model	Full model		
	Proportion mediated, %	Proportion mediated, %	Proportion mediated, %	Proportion mediated, %
Smoking, no. of cigarettes/daycg08106661	26.80	0.038	28.50	0.036
Diet score (ranging 0-6)cg21210537	3.89	0.256	4.66	0.206
cg10643850	-6.73	0.083	-6.91	0.088
cg05740632	11.03	0.062	11.30	0.068
Body mass index, kg/m^2^cg20302171	-2.90	0.269	-2.87	0.267
cg08484100	-2.32	0.304	-1.91	0.373
Systolic blood pressure*, mmHg cg23398826	13.21	0.002	7.65	0.003
cg13311494	3.14	0.082	15.61	0.031
Diastolic blood pressure*, mmHg cg23398826	16.32	0.002	6.39	0.006
cg13311494	3.80	0.079	12.38	0.045
Total cholesterol, mmol/L cg26334131	-5.67	0.449	-31.62	0.168
cg05740632	-33.04	0.020	-3.21	0.126
cg21210537	-6.70	0.169	-8.19	0.197
cg19583211	8.67	0.114	2.73	0.270
Cholesterol in LDL, mmol/L cg26334131	-3.25	0.406	-32.14	0.135
cg20302171	-4.47	0.263	-10.48	0.161
cg05740632	-17.77	0.032	-3.38	0.110
cg19583211	9.03	0.175	3.60	0.210
cg13311494	9.02	0.177	3.70	0.208
cg21210537	-5.50	0.175	-8.55	0.177
Cholesterol in HDL, mmol/L cg15833447	-3.69	0.293	-10.77	0.180
cg21210537	3.79	0.260	7.30	0.235
Random blood glucose‡, mmol/L cg10400937	8.92	0.056	6.72	0.107
cg01545454	6.08	0.122	9.58	0.097
cg11754670	1.26	0.613	36.39	0.086
cg26334131	1.50	0.612	32.70	0.109
cg07219103	6.76	0.102	6.17	0.135
cg20302171	2.49	0.397	15.77	0.123

* We added 15 and 10 mmHg to the measured systolic blood pressure and diastolic blood pressure respectively among participants who reported usage of blood pressure-lowering medications.

‡ Additionally adjusted for treatment of diabetes (yes or no) at baseline.

5) Please check the number of incident cases and controls so the numbers are consistent in the abstract, figure 1, introduction, results etc. The numbers as currently shown vary slightly from 489 to 494.

Baseline DNA methylation was measured for 494 CHD cases and 494 matched controls. In the quality control process, we excluded sex mixed-up samples (n=2); samples with missing rate >0.01 across probes (n=2); samples measured in a distinct study batch (n=2). A total of 491 cases and 491 matched controls were retained for the single marker test. Two samples were further excluded during the network analysis because they were outliers during the sample clustering step, with 491 cases and 489 controls retained. We have revised the manuscript to avoid confusion (Line 85-86, and line 114-116).

6) In the first paragraph of the Results section please expand as the readers won't all look at the methodology section. See also (2) above.

Suggest – 491 cases free of CHD at baseline and developing CHD during follow up and 491 controls free of CHD at baseline and follow up and matched for age, gender, region and timing of blood sampling…

We thank the reviewer for the thoughtful comment. We have revised the manuscript to make it easy to follow (Line 81-88).

7) Table 1. Please show statistical significance p value for prevalent hypertension and diabetes and for lipids

We have added the corresponding p values to Table 1.

8) line 122, can you be specific about items related "healthy lifestyle" and are they known risk factors for CHD?

In our study, we found CHD cases were more likely to be daily smokers, have unhealthy dietary habits, and have higher BMI. We have revised the “Results” section to make it clear (Line 92, 93).

9) line 126, in my opinion, the genomic inflation factor is not helpful here as it is not a good indicator for EWAS. It is because CpGs are more much correlated than SNPs, and the inflation factor is very much dependent on the trait.

We showed the inflation factor for comparison with previous studies.^12–16^ The QQ plot was also shown in Supplementary file 2B. No evidence for inflation was observed in the QQ plots. We have revised the manuscript to address this comment (Line 98, 246). If the reviewer and editor think it is redundant to present the inflation factor, we would be happy to remove it.

10) Line 130: The difference "-0.003" between cases and controls is very small and hard to detect. Can you also show the SD of the two CpGs, or simply plot their distributions in cases and controls.

We have added a column to Table 2 to show the SD of each CpG. We also showed the SD of the top two hits in the “Result” section (Line 104, 105, and 108).

11) Line 144: Why do you use "probe" not "CpG" in this paragraph?

We thank the reviewer for pointing this out. We performed a gene enrichment analysis of the annotated genes of CpG sites from the CHD-associated module. We have revised to keep consistent and avoid confusion (Line 122, 127).

12) Line 156 change none to no

We have revised “none” to “no” (Line 159).

13) Line 242: Did you compare the maximum follow-up time and age distribution in the other two prospective publications? I believe the choice of different follow-up time can be an important factor as we are not sure how long it takes CpGs to have effects on CHD.

We have summarized the mean age and mean follow-up time of our study and two previous prospective studies as below. Our study included younger participants and followed them up for a shorter time period. We agree with the reviewer that different follow-up time and age distribution might explain the differences between our findings and previous prospective publications. We have revised the “Discussion section” to address this comment (Line 223, 227, Author response table 4).

**Author response table 4. sa2table4:** 

	Present study	Guarrera S, et al.^13^	Agha Golareh, et al.^1^
No. of cases	491	292	1,895
Sample size	982	584	11,461
Mean age (years)	50.1	52.4	64
Mean Follow-up time (years)	7.6	12.9	11.2

14) Line 312: "lab staff were blinded" does it also mean the cases and controls were randomized on arrays/batches. This is important as we don't want to lose study power by adjusting batch effect.

The cases and controls were not strictly randomized on arrays. However, the lab staff was really blinded to the case/control status. We agree with the reviewer that adjusting the batch effect may lose power to some extent. We have added it as a limitation to address this comment (Line 246-248, 292, and 293).

15) Line 365: Remove the repeated sentence "A total of 56 SVs were generated."

We thank the reviewer for pointing this out. We have deleted (Line 346).

16) Line 434: How were cellular compositions controlled when you could not use SV here?

In the association analyses of 25 CHD-associated CpGs and cardiovascular risk factors, we also have performed smartSVA for each trait. Adjustment for all SVs instead of batch did not change the association materially (Table 4–Source Data 1-7). To further address this comment, we also additionally adjusted for cellular compositions. The results were generally similar (Author response table 5) . If the reviewer and the editors think it is better to present the results with adjustment for cellular compositions, we would be happy to update all tables in the manuscript.

**Author response table 5. sa2table5:** 

	Model 1		+adjusted for CC	
	Effect size(SE)	P	Effect size(SE)	P
Smoking, no. of cigarettes/day cg08106661	1.50E-04 (4.67E-05)	0.001	9.00E-05 (4.39E-05)	0.041
Systolic blood pressure, mmHg cg23398826	-313.75 (91.72)	<0.001	-255.60 (94.88)	0.007
cg13311494	-108.86 (48.16)	0.020	-101.84 (47.02)	0.031
Diastolic blood pressure, mmHg cg23398826	-184.53 (52.74)	<0.001	-151.73 (55.07)	0.006
cg13311494	-59.54 (27.30)	0.029	-54.51 (27.30)	0.046

17) Can you also report the corresponding p-value of the FDR 0.05 threshold?

The corresponding p-value of the FDR = 0.05 threshold was 2.01E-07. We have added to report in the “Result” section (Line 101-102).

18) If you used β-values of DNA methylation in the analysis, please state it.

Yes, we used the β-values of DNA methylation in the analysis. We have added to clarify (Line 334).References

1. Agha Golareh, Mendelson Michael M., Ward-Caviness Cavin K., et al., Blood Leukocyte DNA Methylation Predicts Risk of Future Myocardial Infarction and Coronary Heart Disease. Circulation 2019;140(8):645–57.

2. Langfelder P, Horvath S. WGCNA: an R package for weighted correlation network analysis. BMC Bioinformatics 2008;9(1):559.

3. Koch A, Joosten SC, Feng Z, et al., Analysis of DNA methylation in cancer: location revisited. Nat Rev Clin Oncol 2018;15(7):459–66.

4. Martínez-Iglesias O, Carrera I, Carril JC, Fernández-Novoa L, Cacabelos N, Cacabelos R. DNA Methylation in Neurodegenerative and Cerebrovascular Disorders. Int J Mol Sci 2020;21(6):2220.

5. Li X-G, Ma N, Wang B, et al., The impact of P2Y12 promoter DNA methylation on the recurrence of ischemic events in Chinese patients with ischemic cerebrovascular disease. Sci Rep 2016;6(1):34570.

6. Leek JT, Johnson WE, Parker HS, Jaffe AE, Storey JD. The sva package for removing batch effects and other unwanted variation in high-throughput experiments. Bioinforma Oxf Engl 2012;28(6):882–3.

7. Nakatochi M, Ichihara S, Yamamoto K, et al., Epigenome-wide association of myocardial infarction with DNA methylation sites at loci related to cardiovascular disease. Clin Epigenetics 2017;9:54.

8. Guarrera S, Fiorito G, Onland-Moret NC, et al., Gene-specific DNA methylation profiles and LINE-1 hypomethylation are associated with myocardial infarction risk. Clin Epigenetics 2015;7:133.

9. Li J, Zhu X, Yu K, et al., Genome-Wide Analysis of DNA Methylation and Acute Coronary Syndrome. Circ Res 2017;120(11):1754–67.

10. Chambers JC, Loh M, Lehne B, et al., Epigenome-wide association of DNA methylation markers in peripheral blood from Indian Asians and Europeans with incident type 2 diabetes: a nested case-control study. Lancet Diabetes Endocrinol 2015;3(7):526–34.

11. Wahl S, Drong A, Lehne B, et al., Epigenome-wide association study of body mass index, and the adverse outcomes of adiposity. Nature 2017;541(7635):81–6.

12. Nakatochi M, Ichihara S, Yamamoto K, et al., Epigenome-wide association of myocardial infarction with DNA methylation sites at loci related to cardiovascular disease. Clin Epigenetics 2017;9:54.

13. Guarrera S, Fiorito G, Onland-Moret NC, et al., Gene-specific DNA methylation profiles and LINE-1 hypomethylation are associated with myocardial infarction risk. Clin Epigenetics 2015;7:133.

14. Li J, Zhu X, Yu K, et al., Genome-Wide Analysis of DNA Methylation and Acute Coronary Syndrome. Circ Res 2017;120(11):1754–67.

15. Fernández-Sanlés A, Sayols-Baixeras S, Curcio S, Subirana I, Marrugat J, Elosua R. DNA Methylation and Age-Independent Cardiovascular Risk, an Epigenome-Wide Approach: The REGICOR Study (REgistre GIroní del COR). Arterioscler Thromb Vasc Biol 2018;38(3):645–52.

16. Rask-Andersen M, Martinsson D, Ahsan M, et al., Epigenome-wide association study reveals differential DNA methylation in individuals with a history of myocardial infarction. Hum Mol Genet 2016;25(21):4739–48.